# Ultrasound in the Differential Diagnosis of Medial Epicondylalgia and Medial Elbow Pain—Imaging Findings and Narrative Literature Review

**DOI:** 10.3390/healthcare10081529

**Published:** 2022-08-13

**Authors:** Wojciech Konarski, Tomasz Poboży, Andrzej Kotela, Martyna Hordowicz, Kamil Poboży

**Affiliations:** 1Department of Orthopedic Surgery, Ciechanów Hospital, 06-400 Ciechanów, Poland; 2Faculty of Medicine, Collegium Medicum, Cardinal Stefan Wyszynski University in Warsaw, 01-815 Warszawa, Poland; 33rd Department of General Psychiatry, Independent Public Regional Mental Health Care Facility Complex of Dr. Barbara Borzym, 26-600 Radom, Poland; 4Faculty of Medicine, Medical University of Warsaw, 01-938 Warsaw, Poland

**Keywords:** elbow, ultrasound, differential diagnosis, golfer’s elbow, epicondylitis

## Abstract

Medial epicondylalgia (ME), commonly known as “the golfer’s elbow”, typically develops in individuals who perform repetitive forearm movements and weight-bearing activities. It accounts for up to 20% of all epicondylitis cases and is most prevalent in particular sports and occupations. Though the diagnosis can be made based on sole clinical examination, additional imaging might be essential for confirmation of medial epicondylitis and excluding other pathologies of the medial epicondyle region. US imaging, with a sensitivity and specificity of 95% and 92%, respectively, remains a practical and accessible alternative to MRI. However, its diagnostic efficacy highly depends on the operator’s experience and requires proper technique. This article describes the ultrasound examination and technique for adequate visualization of elbow joint structures. It also discusses the differential diagnosis of other common and less-known pathologies of the medial compartment of the elbow, including snapping triceps, medial collateral ligament injury, and cubital tunnel neuropathy.

## 1. Introduction

Epicondylalgia remains one of the most common elbow disorders. It occurs mainly when specific movements, namely flexion and extension, followed by pronation and supination of the elbow, are repeated over a long period [1]. Excessive joint strain over time results in micro-trauma accumulation at the tendon attachment. This, in turn, leads to local tenderness, swelling, and ulnar nerve irritation, which results in pain and discomfort with activity, but it may also be present at rest [2].

The first literature report on epicondylalgia was published in 1882. The condition was then described as ‘lawn-tennis arm’ because the disorder was first observed in tennis players [3]. Since that time, knowledge about this entity has increased, and currently, it is known that epicondylalgia may involve either medial or lateral epicondyle. Lateral epicondylalgia (LE), also known as ‘the tennis elbow’, results from injury to the attachment of extensor carpi radialis brevis but may extend to other tendons. In turn, medial epicondylalgia (ME), called ‘the golfer’s elbow’, involves the muscles originating on the medial epicondyle of the humerus [4]. Although their common names suggest a close relationship with particular types of sport, they are also observed in many people who perform other activities involving repetitive hand movements [3,4]. 

Though ME is mainly diagnosed during routine clinical examination, it should be confirmed using imaging modalities. New imaging techniques, especially magnetic resonance (MRI), were propagated to diagnose elbow pathologies in recent decades. For chronic elbow pain, MRI became the first-choice imaging modality [5]. Typically, MRI allows for establishing a proper diagnosis [5]. However, it does not enable real-time dynamic examination of the joint [4]. Ultrasound (US) offers such a possibility, which is gaining popularity in osteoarticular diagnostic and therapeutic purposes [6].

Contrary to the MRI, US examination is more accessible and less expensive. In the case of a well-trained orthopedic surgery specialist with a solid knowledge of normal anatomy, this imaging tool might be non-inferior to MRI and other currently available imaging modalities [6,7,8]. Notwithstanding, modern technical solutions, including harmonic and spatial compound imaging, have brought significant noise and artifact reduction improvements, contributing to an increase in image quality. Recently introduced phase-inversion techniques also improve the field’s depth and signal-to-noise ratio. That together with technological advances in ultrasound probes allows for image quality comparable to that of an MRI [7,9]. 

This paper aims to describe the US anatomy of the elbow, the diagnostic standard for ME examination, and other clinical entities that need to be considered in the differential diagnosis during the US examination. 

## 2. Materials and Methods

We also conducted a narrative literature review to summarize available knowledge about ME diagnosis and management. Studies and papers describing ME and other clinical entities that might develop near the medial epicondyle were included. We excluded papers discussing pathologies of regions other than elbows or papers that did not concern humans. We prepared a narrative synthesis of relevant papers identified in the *PubMed* database. The following set of keywords was used: tennis elbow, medial epicondylitis, risk factors, ultrasound, doppler, differential diagnosis, and epidemiology. We identified a total of 887 results. Out of them, 103 were excluded (language other than English, Spanish, or Polish). Among the remaining 784, 12 were excluded because they were animal studies. In summary, 65 papers regarding ME and its differential diagnosis were included. In papers suitable for inclusion in the review, we also screened the references to identify relevant papers.

US images from our clinical practice were used to compile a pictorial essay to illustrate different pathologies of the medial epicondylar region. Written informed consent was obtained from patients to use US images for educational and publication purposes. 

## 3. Results

### 3.1. Epidemiology of Medial Epicondylalgia 

ME is 7 to 10 times less prevalent than LE, accounting for 10–20% of all cases of epicondylalgia at most [10,11]. In the general population, the prevalence of ME is less than 1%; however, the disease may affect even 8% of patients in certain occupations, which involve repetitive and forceful activities involving the elbow, e.g., meat cutters and butchers [12,13,14,15,16,17]. The annual incidence of ME is estimated at 0.8–5.6/1000 person-years [14,16]. Although the syndrome has been identified in patients from 12 to 80 years old, it predominantly occurs in the fourth and fifth decades of life [14,17]. 

ME occurs more often in persons with other work-related upper-limb disorders, particularly shoulder tendinitis, LE, and carpal tunnel syndrome [13]. Among non-movement-related factors, age 40 or older, smoking, obesity, and psychological distress were associated with ME development [14,16]. There is no consensus in the literature regarding gender as a risk factor for ME. Some studies report that males and females are affected equally [14,17]. Others indicate that ME occurs twice as frequently in women [15,16]. 

### 3.2. Causes and Other Risk Factors

The leading cause of ME is specific, repetitive arm movements, which result in recurrent trauma to the tendons attached to the medial compartment of the elbow, mostly-extensor carpi radialis brevis. Although the disorder is commonly known as ‘the golfer’s elbow’, it does not occur only in golf players but also in other athletes who participate in activities that stress the wrist and elbow joints, e.g., rowing and baseball, tennis, or bowling. Of note, 90% of ME cases are not sport-related but are diagnosed in some specific occupations [13,14,15,16,17]. Certain movements that may lead to the medial elbow compartment tendinopathy were identified. These include extreme flexion of the elbow, posture with extended elbows, posture with excessive pronation or supination of the elbow, highly repetitive movements of the elbow, grasping or lifting of objects with high forces, and a combination of repetitive postures and movements [18]. Other papers on this topic mention similar factors [10,13,14,19]. In a systematic review by van Rijn et al., ME was found to be associated with handling objects greater than 5 kg for 2 h per day, objects greater than 10 kg more than ten times per day, and repetitive movements and vibrating tools for more than 2 h per day [20]. In a population-based study, even bending or straightening the elbow for more than one hour a day was related to a higher risk of ME [16]. Hence, the disorders occur among carpenters, bricklayers, hammermen, painters, and other physical workers and among persons who use keyboards for typing for extended periods [21].

Nonetheless, some authors argue that several mechanisms might lead to ME development, implying that ME might be found in patients without typical risk factors [13,14]. Shiri et al. suggested that forceful but not repetitive activities and awkward posture are more likely to cause such injury [14]. According to Descatha et al., forceful work was even more closely related to the risk of ME than repetitive work [13]. Some ME cases are due to a single traumatic event such as a violent contraction of the extensors when attempting extremal exertion or a twisting injury [17].

#### Symptoms of Medial Epicondylalgia 

The main symptoms of ME are pain and tenderness. Although ME does not cause severe disability, it might cause debilitating pain interfering with basic everyday activities. The pain can develop suddenly or gradually [22]. It can be of an intermittent or persistent character and is typically localized at the inner side of the elbow, at the muscle–tendon junction, or at the insertion points of the wrist flexors in the elbow region [22,23]. Sometimes, the pain radiates distally to the forearm. It typically worsens with specific movements such as forearm pronation, gripping, or throwing but may also occur at rest, especially in the acute phase [18,22,23,24]. Symptoms are often provoked by lifting heavy objects [2,18].

Initially, the range of motion can be full, but in chronic cases, limitation of wrist extension and a flexion contracture may occur [25,26]. Numbness or tingling may radiate into one or more fingers (usually the ring and little fingers) mainly when concomitant ulnar neuropathy exists [18,19]. Other symptoms include joint stiffness and decreased grip strength. Chronic cases might also be characterized by reduced grip strength [22].

About three-quarters of the patients develop ME in their dominant arms, but some research outcomes indicate no association between the side affected by ME and the worker’s handedness [13,17].

### 3.3. Pathophysiology of Medial Epicondylalgia

The cause of ME is an injury to the insertion of the pronator-flexor muscle group on the medial epicondyle of the humeral bone [22,25]. Initially, the inflammatory process was considered to play the primary role in ME development [27]. The term suggesting an inflammation is still being used; however, histologically, the disease is caused by the accumulation of microtraumas over time, caused by supraphysiologic stress on the tendon. The trauma results in angiofibroblastic degradation, fibrosis, and calcification of the fibrous structures at the medial epicondyle [28]. Vascular and fibroblastic elements replace the normal tendon; during that process, mucoid degenerates, and reactive granulation tissue is formed. The failed reparative process leads to fibrosis or calcification, which decreases collagen strength through scar tissue formation and thickening of the tendons [21,29,30,31]. Therefore, some specialists prefer the term ‘tendinosis’ or ‘tendinopathy’ when describing elbow epicondylalgia [14]. In summary, though ‘medial epicondylitis’ is still being used to name this pathology, the term ‘medial epicondylalgia’ is probably more suitable as it avoids the usage of the -itis suffix, which suggests an inflammatory background of this condition.

In most cases, ME usually develops gradually. Because pain subsides with rest, most patients would not seek treatment early. Therefore, the diagnosis is usually made at the chronic stage when fibrotic degradation has already begun. When the muscle attachment is affected, its weakened structure becomes more susceptible to further microtrauma [22,23]. Damage to the structure of the attachment increases if the affected limb is subjected to further long-lasting and unlimited activity. However, some ME cases result from acute trauma resulting from a sudden contraction of the muscles attached to the medial epicondyle [14].

### 3.4. Diagnosis

#### 3.4.1. Normal Anatomy of the Medial Epicondyle Region

The epicondyles are rounded bony protuberances at the distal end of the humerus. The medial epicondyle is located inside the elbow on the humerus. It is the attachment site for five muscles and tendons, which form the common flexor tendon (CFT). This musculotendonous structure includes (from proximal to distal) the pronator teres (PT), the flexor carpi radialis (FCR), the palmaris longus (PL), the flexor digitorum superficialis (FDS), and the flexor carpi ulnaris (FCU). This tendinous structure forms the deepest part of the medial ligament complex and plays a vital role in stabilizing the joint medially [22,23,32]. Figure 1 provides an anatomical overview of the muscles attached to the medial epicondyle.

The elbow joint performs the movements within the specified range: flexion 130–140°, extension 180°, pronation 60–80°, and supination 70–85° [32]. However, chronic repetition of forearm pronation and wrist flexion leads to disorders that may involve almost all CFT (except for palmaris longus). Still, the pronator teres and the flexor carpi radialis were previously thought to be most affected [17,23]. During US examination of the medial epicondyle, physiological attachment of the CFT to the medial epicondyle of the humerus is presented in Figure 2. 

#### 3.4.2. Clinical Diagnosis

The ‘golfer’s elbow’ diagnosis is primarily based on clinical signs and symptoms. Nonetheless, it requires a careful physical examination and interview. In selected cases with an atypical presentation, imaging might be necessary to differentiate between ME and rule out other possible causes of medial elbow pain [17,33].

The medical interview should be focused primarily on the patient’s history. It is vital to establish if the patient has a history of activities involving either repetitive elbow use, gripping, valgus stress, or acute trauma [34]. About 30% of cases are associated with an acute injury, whereas 70% have a more insidious onset. Information on symptoms’ timing and duration must also be obtained [5,22].

Typical pain in ME is intermittent, directly located in the medial epicondyle region, and is activity-dependent. Tenderness at the insertion of the flexor-pronator muscle complex (the CFT), situated 0.5–1 cm distally from the medial epicondyle, is characteristic of ME. Symptoms should be present at the time of testing or occur at least four days during the last seven days. A typical finding is local pain during the ME test during clinical evaluation [5,34]. It is performed with the patient’s elbow extended and fully supinated. The examiner puts one hand on the patient’s ventral side of the hand, stabilizes the elbow with the other hand, and asks the patient to move his hand to palmar flexion against resistance. Pain during the resisted wrist flexion and pronation is the most sensitive test for ME during the physical examination [5,18,35,36,37]. On physical examination, swelling, erythema, or warmth may be present in rare acute cases, but most patients present with chronic ME with no apparent signs of inflammation [5,36]. 

The difference in upper limb range of motion, especially during forearm pronation, or a decrease in forceful grip compared to the contralateral side might also be observed. These movements might also be accompanied by elbow pain [29,37,38]. When examining overhead athletes, it is critical to evaluate for ulnar neuritis and ulnar collateral ligament instability, which may also coexist [37]. A provocative test to aid in the diagnosis is an exacerbation of pain with resisted forearm pronation with wrist extension [36,37,38].

In addition to the above case definition criteria and accompanying symptoms, Polk’s tests may be employed to assess for ME. During this test, the patient sits with his elbow flexed about 100°, and his forearm supinated. The examiner asks the patient to grab and lift the object weighing approx. 2.5 kg. Pain in the medial epicondyle indicates that the test is positive [36,39,40].

#### 3.4.3. Imaging of Medial Epicondylalgia

Typically, imaging using US, MRI, or other imaging modalities allows for distinguishing between different pathologies of the medial elbow [22]. 

#### 3.4.4. Radiography

Plain radiographs of the elbow in ME patients are usually normal. However, it may demonstrate sclerotic changes in chronic cases and collateral ligament calcification in throwing athletes. A lateral X-ray is also helpful in eliminating alternate diagnoses, including medial epicondyle fracture, elbow arthritis, and deformity [19,23].

#### 3.4.5. Magnetic Resonance Imaging 

Magnetic resonance imaging (MRI) without contrast appears to be the method of choice for the radiologic evaluation of ME and other causes of pain in the elbow joint [5]. Usually, it is performed if the clinical picture of the disease is unclear. On T1- and T2-weighted sequences, the thickening of the CFT (from intermediate to high) indicates ME [41]. The most characteristic finding is increased signal intensity on the T2-weighted images in the CFT area and paratendinous edema [40]. MRI is also practical for detecting pathologic changes in tendons, including MCL [10]. 

#### 3.4.6. Ultrasonography 

Although MRI is regarded as a gold standard for confirming the ME, US imaging, with a sensitivity and specificity of 95% and 92%, respectively, remains a practical and accessible alternative. It was demonstrated that a sonogram performed by a trained professional has positive and negative predictive values of over 90% for ME diagnosis. Still, it is noteworthy that the diagnostic efficacy of ultrasonography is highly dependent on the operator’s experience [10,23,42]. Ultrasound can also be used for therapeutic purposes during guided injections. It might also serve as a tool for assessing the response to treatment [37,43,44].

Proper imaging is obtained when the tested limb is in 90 degrees flexion, and the forearm is in intermediate rotation. The probe should be placed along the long axis of the forearm to lie in the proximity of the humeral medial epicondyle. By rotating the probe to 90 degrees, a cross-section can be visualized. A normal flexor attachment has a typical fibrous structure, intermediate echogenicity, and uniform thickness. In patients with ME, US may show thickening and heterogeneity of the common extensor tendon and hypoechoic or anechoic areas of focal tendon degeneration (Figure 3) [37,42,43]. More advanced tendinopathy, ruptures, and evidence of calcification of the CFT are showed in Figure 6 and Figure 7 or MCL may be observed. 

New ultrasound technologies have also found an application in assessing elbow pathologies. Ultrasound elastography (USE), which was first implemented in the 1990s, is increasingly being used to determine tissue stiffness quantitatively. The elasticity of tissues is often altered because of pathologic processes, such as fibrosis. Therefore, USE has proven helpful for evaluating hepatic cirrhosis in oncology and orthopedics [45,46]. This technique may be helpful in epicondylalgia diagnosis; hence, increased compressibility is characteristic of this disorder [45]. Shear wave elastography (SWE) and strain elastography (SE) were recently similar diagnostic utility in ME [46]. However, USE as a routine test requires further research and standardization of the technique. 

Recently, power Doppler use has emerged as a tool with high diagnostic accuracy in LE [47]. It allows for visualizing hyperemia, which accompanies inflammatory processes [48,49]. Like USE, the power Doppler utility in ME diagnosis requires further research.

##### The Technique of Ultrasound Examination of the Medial Portion of the Elbow

As the joint flexes or extends during the US examination, a change in the shape of the medial collateral ligament (MCL) might be observed. The probe should be placed right above the medial epicondyle while the patient flexes and extends the forearm. The normal appearance of the MCL is depicted in Figure 4.

The CFT should be examined in longitudinal and transverse sections during the US examination. It consists of the superficial fibrous part, which is hypoechogenic, and the deep part, which is moderately hyperechogenic (Figure 4c). Near the CFT lies the MCL, which is best observed on the long axis view (Figure 4c).

Another structure that must be assessed during the US examination of the ME is the ulnar nerve. The ulnar nerve is best visualized when the probe is moved medially from the olecranon towards the ME [50]. Depending on whether the ulnar nerve is also affected, ME is classified into two subtypes: without (type 1) and with (type 2) ulnar nerve involvement [1,23]. The nerve lies in the elbow’s posterior part and the ulnar nerve groove. Moving distally, the ulnar nerve penetrates the forearm muscles through the ulnar and humeral FCU heads. In ulnar nerve neuropathy, the nerve is frequently compressed at the cubital tunnel retinaculum, a thick membrane called the Osborne’s band [50]. The latter lies superficially to the ulnar nerve. Examining the individual variations in the ulnar nerve anatomy is vital before therapeutic injections in CFT enthesopathy. In addition, during flexion and extension movements, subluxation and dislocation of the ulnar nerve might be observed. The technique for US examination of the ulnar nerve is demonstrated in Figure 5.

The US picture of ME enthesopathy is presented in Figure 6, Figure 7 and Figure 8.

**Figure 6 healthcare-10-01529-f006:**
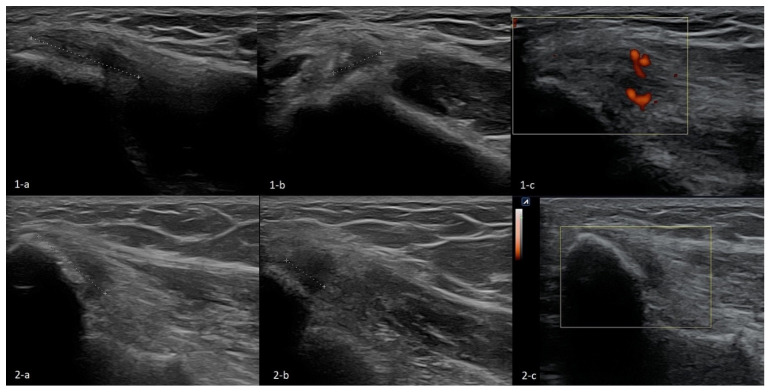
Two cases of simple enthesopathy. Each series contains an image from the longitudinal view (**a**), transverse view (**b**), and power Doppler mode long-axis view (**c**). In images **1-a**–**1-c**, the superficial part of the attachment CFT is involved, and the power Doppler test shows moderately increased vascularization, probably indicative of the regenerative processes. In the second case (images **2-a**–**2-c**), the superficial and deep part of the attachment is involved; no signs of increased vascularization are found in the power Doppler mode.

**Figure 7 healthcare-10-01529-f007:**
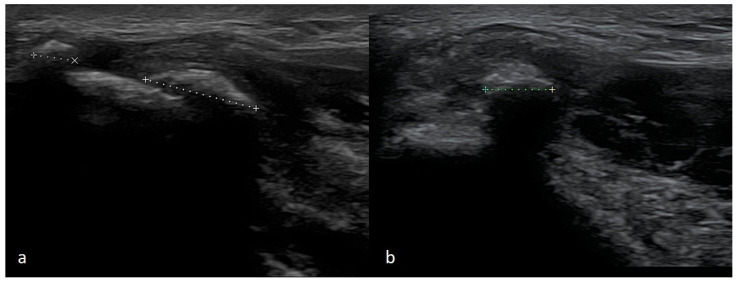
A case of advanced CFT enthesopathy with calcifications against damage within the flexor’s attachment structure. (**a**)—Long-axis view, (**b**)—short-axis view.

**Figure 8 healthcare-10-01529-f008:**
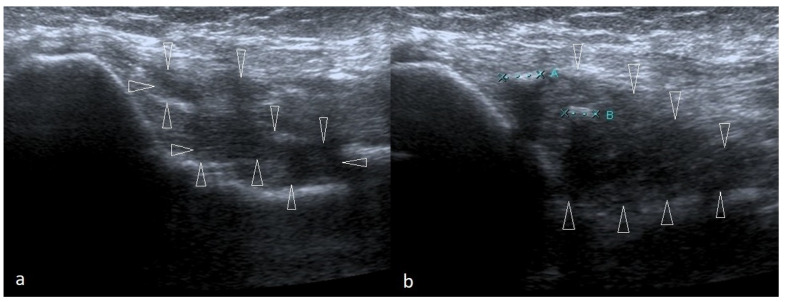
Two cases of post-traumatic changes within the medial collateral ligament and profound part of the common flexor tendon. In both pictures (**a**,**b**), the areas of hypoechoic scarring are marked with arrowheads; post-traumatic calcifications are also visible.

## 4. Discussion

ME diagnosis may be based on the physical examination and the patient’s medical history. Nonetheless, ME is a relatively uncommon disease with unspecific symptoms, so the entity can easily be confused with other sources of elbow pain—features differentiating ME from other conditions [34,51]. 

### 4.1. Cubital Tunnel Syndrome and Ulnar Neuritis

Cubital tunnel syndrome is the second most common compression neuropathy. The ulnar nerve courses behind the medial epicondyle, so the pressure or stretching in this area may affect the shape of the cubital tunnel, which causes pain [34]. However, contrary to ME, where pain with activity is the leading symptom, most patients report sensory loss first. Numbness, tingling sensation, and weakness of the forearm muscles can also occur [34,52].

Diagnosis of cubital tunnel syndrome may be facilitated with the Tinel test, which involves direct gentle compression over the nerve course at the elbow level. The test is positive when a sensation of tingling and paresthesia occurs [5,10,34,37]. The diagnosis can be excluded by performing electromyography (EMG). It might also elucidate the site of nerve compression and the degree of nerve damage. Some authors recommend screening patients for systemic and metabolic disorders that might predispose them to this condition [53,54]. At times, the neuropathy might result from nerve instability or subluxation. Nerve displacement occurs mainly during forearm flexion to more than 90 degrees [55]. When the ulnar nerve dislocates from its groove, it becomes irritated, which might be captured on the US as nerve swelling. The dislocation might be observed when examining the nerve with a supinated forearm. A patient is asked to flex the forearm until the full flexion is reached gradually [55].

Nonetheless, ulnar nerve edema might be secondary to forceful work or after long periods of forearm flexion. In such circumstances, the patient might also experience symptoms indicative of neuropathy [55]. The US image of ulnar nerve pathologies is depicted in Figure 9.

### 4.2. Snapping Triceps with Ulnar Neuritis

Snapping triceps as a cause of ulnar neuritis Although it is rarely diagnosed, snapping triceps can also cause ulnar neuritis [56]. It affects men primarily in the fourth decade of life. Dynamic US examination is a method of choice because it allows differentiation between the snapping of the muscle and the ulnar nerve (Figure 10) [57]. The muscle may dislocate both medially and laterally. The treatment for the snapping triceps syndrome is primarily conservative and involves avoiding activities that provoke symptoms.

### 4.3. Ulnar Nerve Compression Caused by Anconeus Epitrochlearis

Another rare cause of ulnar neuritis is the presence of an anomalous muscle, anconeus epitrochlearis (AE). Cadavers and imaging studies found it in 3–34% of subjects [58,59,60]. Nonetheless, this accessory muscle in patients undergoing surgery for cubital tunnel syndrome might be as high as 20% [61]. Typically, patients with AE muscle and ulnar neuritis develop symptoms more rapidly and early in life; forearm movements might sometimes provoke the symptoms.

### 4.4. Ulnar Collateral Ligament Injury

Injury to the medial collateral ligament (MCL) occurs mainly in athletes who throw overhead because the anterior part of the MCL is the primary restraint to valgus stress during overhead throwing. The disorder leads to valgus elbow instability and pop sensation over the medial elbow [5,62,63].

The most crucial test for MCL injury is ‘the valgus stress test’, also known as ‘the elbow abduction stress test’, which involves palpation of the medial joint line of both the symptomatic elbow and the contralateral side and comparing them for signs of laxity or instability against valgus forces. Valgus stress should be applied against an elbow flexed 20–30°, and then, the amount of opening and the subjective quality of the endpoint is assessed. The test is positive when a firm endpoint is absent, joint space opens more than 3 mm, or the patient feels pain [21,51]. MCL injury may also be confirmed with a positive result on the moving valgus stress test or milking maneuver [5,36]. An US picture of MCL is presented in Figure 8.

### 4.5. Cervical Radiculopathy

Patients with neurologic disorders should also be examined for cervical radiculopathy and dysfunction of a nerve root of the cervical spine. The C7 and C6 cervical nerve roots are the most affected. The typical symptoms of cervical radiculopathy are neck and arm discomfort of insidious onset and sensory changes along the involved nerve root dermatome, including tingling, numbness, or sensorineural loss [64]. Confirming cervical radiculopathy may be possible with the foraminal compression test (or Spurling test). It is performed by applying downward pressure on a patient’s head, with the neck extended and the head rotated. The test is positive if the pain radiates into the ipsilateral limb. As C6 and C7 radiculopathy lead to muscle weakness, it may predispose to ME development [64,65]. An imaging modality of choice for confirmation of cervical radiculopathy is MRI.

### 4.6. Ganglion Cyst

Another cause of elbow pain may be a ganglion cyst, a benign form of soft tissue swelling. It typically arises from the ulnohumeral joint capsule. The etiology of ganglia remains unclear, but degenerative changes at the collective and repeated minor trauma seem to contribute to its development [66]. An improper diagnosis is likely because the ganglion may mimic epicondylalgia or cubital tunnel syndrome. US evaluation (Figure 11) or magnetic resonance imaging (MRI) may be particularly helpful in the proper assessment of pain sources [67,68,69].

### 4.7. Degenerative Changes in the Elbow Joint

Osteoarthritis and degenerative changes in the elbow joint might cause elbow pain and stiffness, especially in elderly subjects [70]. The most common cause of elbow arthritis is rheumatoid arthritis, followed by trauma-related arthritis and primary osteoarthritis [71]. Nonetheless, it seems unlikely that such changes will be highly localized to affect only the medial side of the elbow without changes in other joints. US dynamic examination might help determine if osteophytes and loose bodies are a source of irritation and pain and exclude inflammatory activity that might suggest a rheumatic background of the patient’s ailments.

### 4.8. Epitrochlear Lymphadenopathies

Lymphadenopathy of the elbow region is likely to signify a systemic illness. Usually, they accompany the disorders that cause general lymphadenopathy (including lymphomas, leukemias), infections (Epstein–Barr virus, human immunodeficiency virus, cat-scratch disease, syphilis), and malignancies [72,73,74]. Patients with enlarged lymph nodes in the epitrochlear region should be carefully evaluated in the light of clinical data, such as the presence of fever and other systemic symptoms, mobility of enlarged nodes concerning skin and deeper layers, and changes in the overlying skin [73]. It is advisable to screen such patients for red flags indicative of malignant disease and refer them for further testing to exclude such pathology [73].

## 5. Conclusions

Medial epicondylalgia, although less frequent than involvement of lateral epicondyle, should not be underestimated as a cause of elbow pain. Such patients may struggle to perform everyday activities at home and work. Ultrasound examination might be a valuable complementary assessment tool for an orthopedic specialist, which allows for the assessment of both the medial epicondyle and adjacent structures, including the ulnar nerve, muscles, and their attachments. It is also helpful in excluding the dynamic nature of the patient’s symptoms, which occur in both ME, ulnar nerve dislocations, and snapping triceps syndrome, among others.

## Figures and Tables

**Figure 1 healthcare-10-01529-f001:**
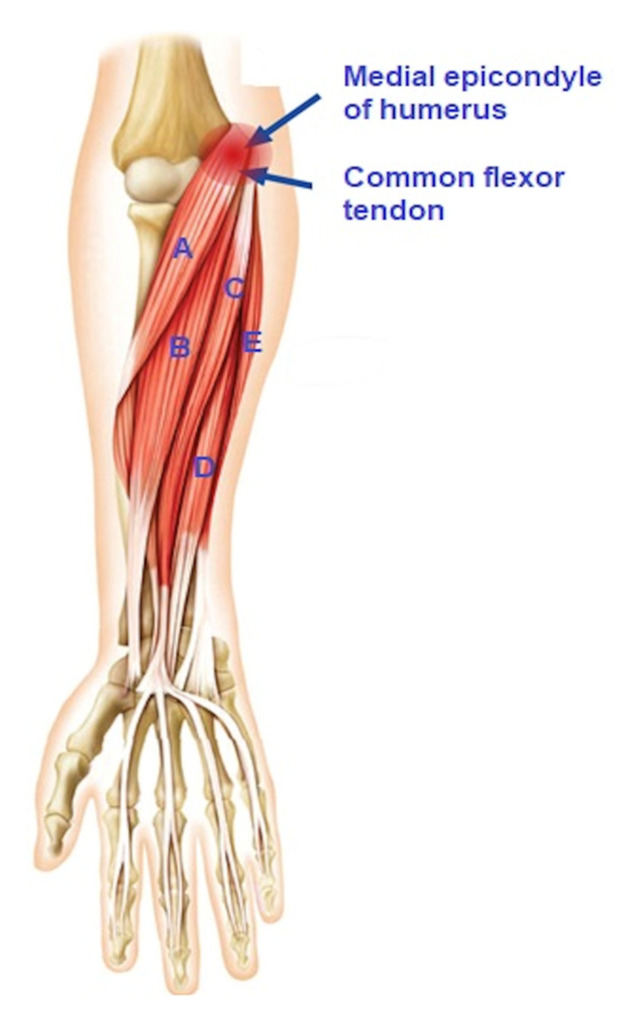
Muscles of the elbow and their attachment of medial epicondyle of humerus. Figure depicting elbow’s muscle attachment. The letters represent: A—pronator teres muscle; B—flexor carpi radialis; C—palmaris longus; D—flexor digitatorum superficialis; E—flexor carpi ulnaris.

**Figure 2 healthcare-10-01529-f002:**
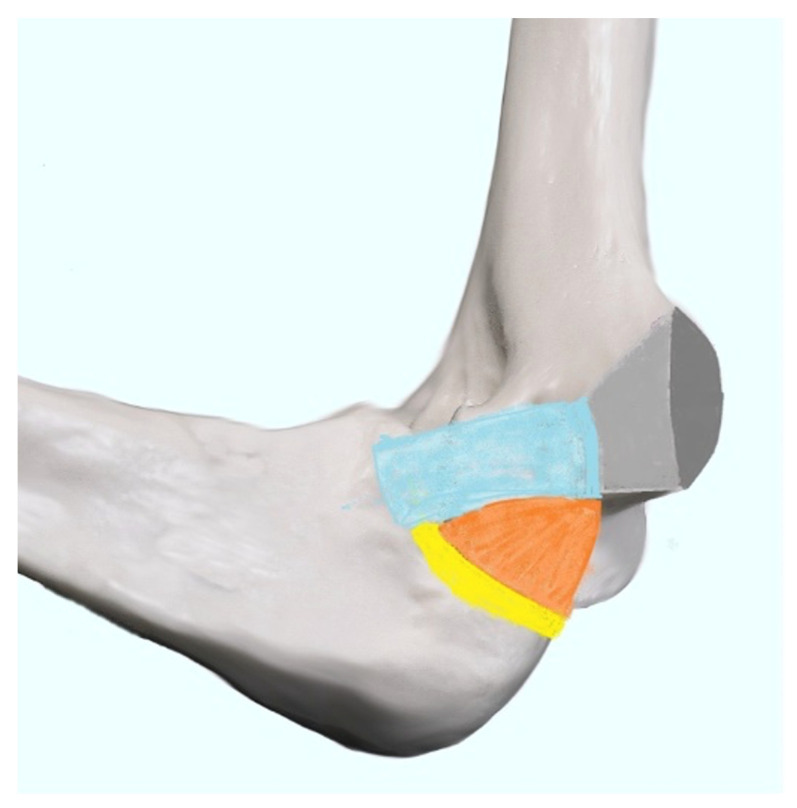
Anatomical diagram showing the site of typical attachment of flexors (CFT) and medial collateral ligament. A dark gray area represents the site of attachment of the superficial, less fibrotic part of the attachment of CFT; a light gray area—the site of attachment of a deeper, less fibrotic part of it; complex, blue—anterior part of the medial collateral ligament; orange—posterior part, with the transverse part’s attachment marked in yellow.

**Figure 3 healthcare-10-01529-f003:**
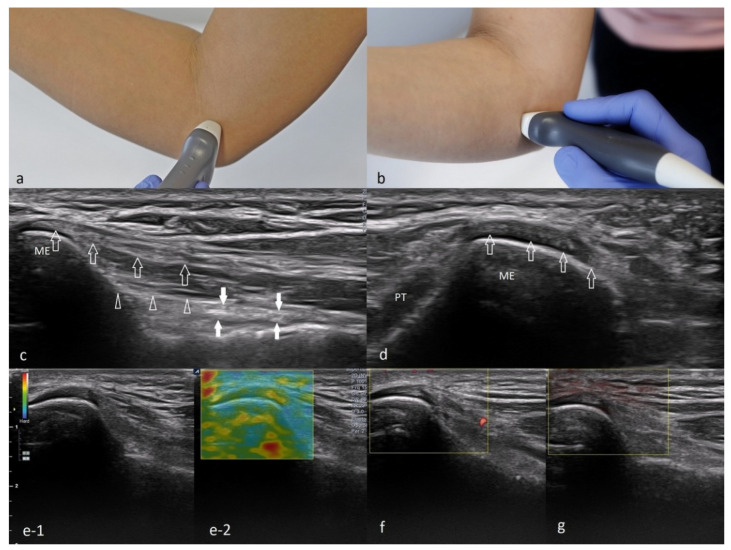
Ultrasound examination of common flexor attachment. Images (**a**,**b**)—a method of applying the probe to obtain both (**a**)—longitudinal, (**b**)—transverse sections, (**c**)—long axis view of a common flexor attachment (precisely above the attachment of the part corresponding to the flexor carpi radialis—FCR); open arrows—superficial, more tendinous portion of the FCR; open arrowheads—deep part of the FCR, with preserved muscular echo structure; white arrows—partially visible anterior part of the medial collateral ligament; ME—medial epicondyle. (**d**)—Short axis view of the common flexor attachment—open arrows; ME—medial epicondyle; PT—proximal part of the pronator teres muscle (the humeral head). (**e**)—Scans from strain elastography, (**f**)—normal CFT in power Doppler mode, (**g**)—normal CFT assessed with MicroFlow option.

**Figure 4 healthcare-10-01529-f004:**
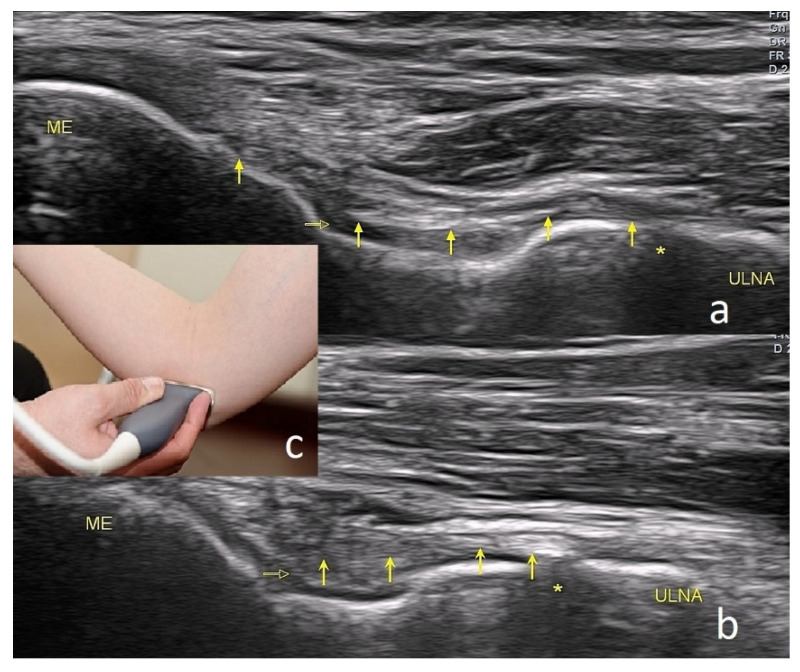
Image of the anterior part of the medial collateral ligament. (**a**)—During flexion, (**b**)—during extension, (**c**)—positioning of the probe; ME—medial epicondyle; arrows—the ligament; open horizontal arrow—joint capsule level; asterisk—the joint gap.

**Figure 5 healthcare-10-01529-f005:**
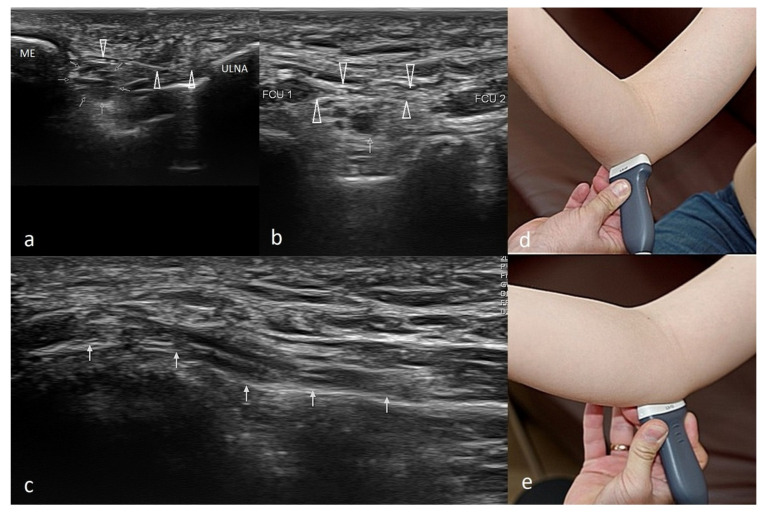
Ultrasound examination of the ulnar nerve. (**a**)—Short-axis view at the level sulcus of the nerve; small arrows—ulnar nerve; open arrowheads—Osborn’s ligament; ME—medial epicondyle. (**b**)—Short-axis view just distal to the medial epicondyle; small arrow—ulnar nerve; FCU1—humeral head of the flexor carpi ulnaris; FCU2—ulnar head of the flexor carpi ulnaris; arrowheads—Osborn’s band connecting proximal parts of both heads of the flexor carpi ulnaris. (**c**)—Long-axis view of the ulnar nerve-arrows; (**d**,**e**)—a method of applying the probe to obtain both (**d**)—transverse and (**e**)—longitudinal sections.

**Figure 9 healthcare-10-01529-f009:**
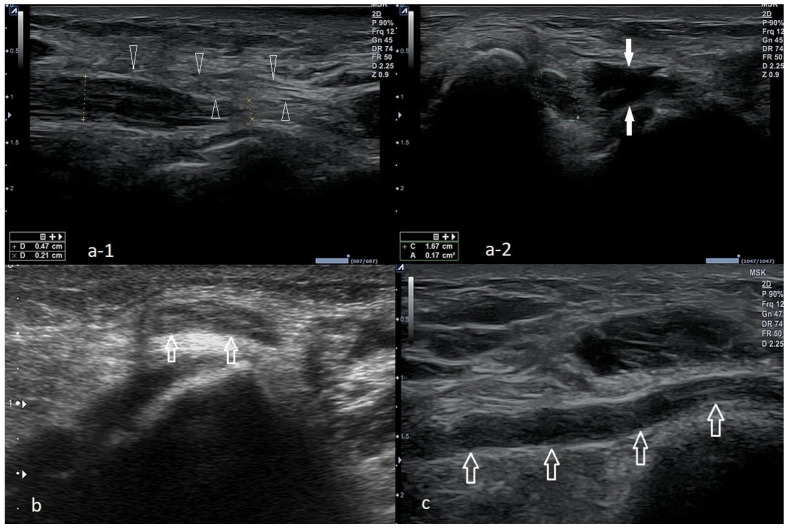
Three cases of the ulnar nerve neuropathy of different etiologies. (**a-1**)—Long-axis view of the ulnar nerve, visible compression of the nerve at the level of the arcuate ligament (the narrowing of the nerve is accompanied by increased echogenicity); proximal to the stenosis site, nerve swelling is visible, expressed as a thickening, decreased echogenicity without the typical echo structure. Arrowheads—Osborn’s ligament transforming into an arcuate ligament in the distal direction. (**a-2**)—The same case, cross-section at the level of the sulcus, significantly increased cross-sectional area (0.17 cm^2^, with the norm defined by most authors at 0.09 cm^2^), although in the longitudinal section, a nerve compression is visible at the level of the arcuate ligament; in this case, at the level of the sulcus there is also a visible muscular structure that may correspond to the anconeus epitrochlearis (marked with white arrows). (**b**)—A case of instability of the ulnar nerve, during flexion above 90 degrees: the ulnar nerve (arrows) was dislocated above the surface of the medial epicondyle. (**c**)—Significant nerve swelling, mainly at the level of the sulcus; no stenosis was visualized at any of the levels.

**Figure 10 healthcare-10-01529-f010:**
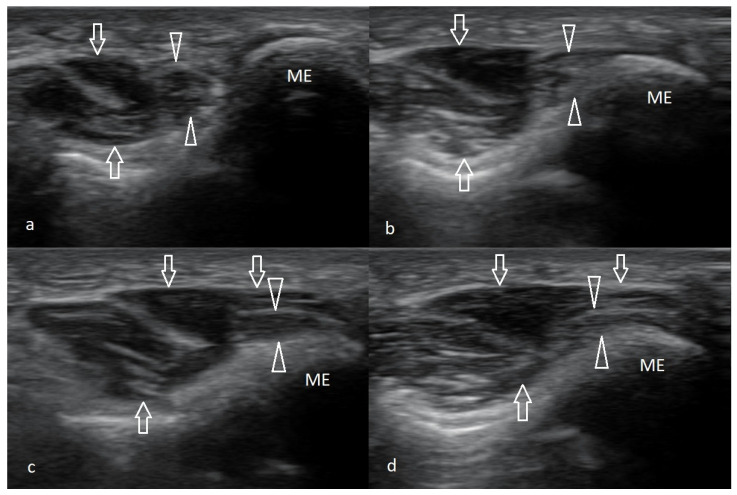
Snapping triceps syndrome. A series demonstrating the anterior dislocation of the medial head of triceps muscle over the surface of the medial epicondyle, which is accompanied by shifting the ulnar nerve beyond the sulcus and its compression; compression increases as the degree of flexion increases. (**a**)—Examination in full extension, (**b**)—90, (**c**)—100, and (**d**)—120 degrees of flexion; arrows—medial head of triceps muscle; arrowheads—ulnar nerve.

**Figure 11 healthcare-10-01529-f011:**
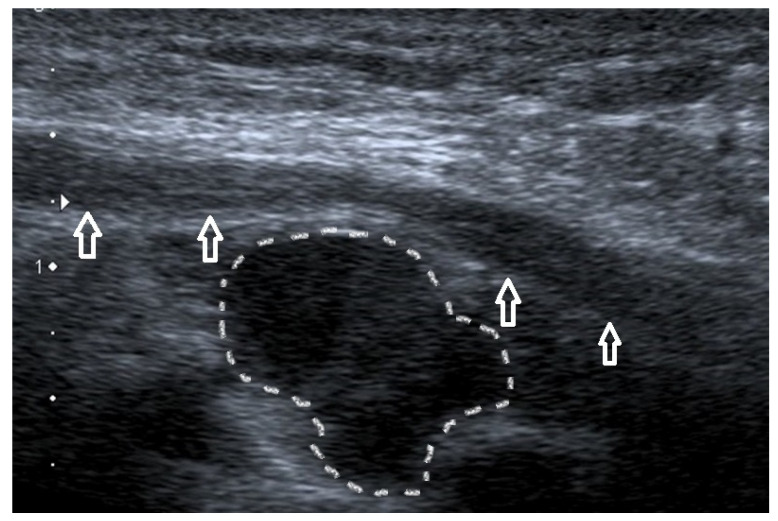
Ganglion cysts (dotted line) compressing the ulnar nerve (arrows).

## Data Availability

Not applicable.

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
