# Peer review of "Ultrasound in the Differential Diagnosis of Medial Epicondylalgia and Medial Elbow Pain—Imaging Findings and Narrative Literature Review"

_healthcare, 2022, doi:10.3390/healthcare10081529_

Round 1

Reviewer 1 Report

The paper is well written with an exhaustive description of the technique and its applications. In well-defined sections, the authors have provided an in-depth outline of the disease, pathology and the application of US - a more economical and widely available imagine resource for ME. A few specific comments are below:

Line 65: Could the authors also mention the exclusion criteria? Were there any conditions of the elbow region where this technique would not be equally suitable?

Line 79: Authors may provide a few examples here.

Line 90: Could the authors include a diagram of the elbow anatomy in order to understand the anatomical terms better?

Paragraph 3.4: Please give a couple of examples here in addition to the references. The sentences seem incomplete without at least a couple of examples.

Line 147: Could the authors give a reason why the diagnosis takes time?

Line 255: Is this line complete?

Author Response

Dear Reviewer, many thanks for your comments. We also believe that the US examination is an underestimated tool in orthopedics compared to the MRI. Our paper aimed to explain the technical aspects of its use as well as provide a summary of the current state of the art about ME, with an emphasis on the clinically relevant aspects.

A few specific comments are below:

Line 65: Could the authors also mention the exclusion criteria? 

Usually, narrative reviews do not have a set of well-defined inclusion and exclusion criteria, and the methodology is less rigorous than in the case of systematic reviews. In this review, we only took into consideration papers regarding elbow pathologies in humans, especially in the medial epicondylar region.

Were there any conditions of the elbow region where this technique would not be equally suitable?

We don’t fully understand what this query refers to. Did you mean our US technique (described in lines 237-245)?

Line 79: Authors may provide a few examples here.

We have added a few examples, but more importantly - types of activities that research demonstrated to carry a significantly higher risk of ME development.

Line 90: Could the authors include a diagram of the elbow anatomy to understand the anatomical terms better?

This is a very reasonable request. We have added a new figure (figure 1) with a description of elbow anatomy.

Paragraph 3.4: Please give a couple of examples here in addition to the references. The sentences seem incomplete without at least a couple of examples. 

We are afraid that we do not fully understand what needs more elaboration. Paragraph 3.4 is „diagnosis,” which is divided into numerous subparagraphs. Please specify to which part/lines you refer.

Line 147: Could the authors give a reason why the diagnosis takes time? 

The diagnostic process itself is not that complicated. Usually, the disease has an insidious onset, with symptoms increasing gradually. This translates into a diagnostic delay. We have included a brief explanation in the text, as you suggested.

Line 255: Is this line complete?

That line was also confusing for another reviewer. It was meant to be an intermediate title that we now decided to delete. 

Reviewer 2 Report

The paper presents a general overview of Medial Epicondylitis (ME), focusing on the diagnostic capacity of ultrasound applied in this and other area-related pain sources. The authors state that ME may not be an inflammatory-mediated condition in many instances. Therefore, I would suggest the term Medial Epicondylalgia be used throughout the manuscript to avoid the –itis suffix.

The authors seem to have a good understanding of the pathology / pathophysiology of medial epicondylitis and ultrasound imaging methods and techniques to identify lesions in the tendons or other structures involved effectively.

Although this is not a systematic review, in the Materials and Methods, in my opinion, the number of references initially identified and how these were grouped, as well as why some were possibly excluded, might be worth mentioning to present the amount of literature dealing with this topic.

In Figure 1, include in the legend the difference between part A & part B of the figure.

Line 244: In the ’90s (correct)

Line 255: Is this an intermediate title?

Lines 286-287: Why is the annular ligament mentioned here? This is a structure of the lateral side of the elbow.

I would have preferred this paper to be a more focused systematic review; however, there is maybe a point to present as is.

Author Response

Dear Reviewer, many thanks for your comments.

A few specific comments are below:

The paper presents a general overview of Medial Epicondylitis (ME), focusing on the diagnostic capacity of ultrasound applied in this and other area-related pain sources. The authors state that ME may not be an inflammatory-mediated condition in many instances. Therefore, I would suggest the term Medial Epicondylalgia be used throughout the manuscript to avoid the –itis suffix.

This is an excellent suggestion. We have adapted your idea in section 3.3 (pathophysiology of ME).

The authors seem to have a good understanding of the pathology / pathophysiology of medial epicondylitis and ultrasound imaging methods and techniques to identify lesions in the tendons or other structures involved effectively.

Although this is not a systematic review, in the Materials and Methods, in my opinion, the number of references initially identified and how these were grouped, as well as why some were possibly excluded, might be worth mentioning to present the amount of literature dealing with this topic.

This is not usually done for narrative reviews, but we have included key numbers in the Methods section of the manuscript to show the amount of literature discussion ME.

In Figure 1, include in the legend the difference between part A & part B of the figure. 

The part ‚a’ of that figure was unnecessary; we only decided to keep the figure ‚b’ and corrected the figure description.

Line 244: In the ’90s (correct)

Thank you. We have corrected that mistake according to your remark.

Line 255: Is this an intermediate title?

That line was also causing confusion in another reviewer. As you said, it was meant to be an intermediate title that we now decided to delete. 

Lines 286-287: Why is the annular ligament mentioned here? This is a structure of the lateral side of the elbow.

We truly apologize for that obvious mistake. We meant „cubical tunnel retinaculum (Osborne’s ligament). That sentence was corrected. We genuinely thank you for noticing it.

I would have preferred this paper to be a more focused systematic review; however, there is maybe a point to present as is.

Many thanks for your suggestion. Systematic reviews (with or without a meta-analysis) became a „golden standard” in data synthesis. Nonetheless, narrative reviews bear several advantages over strict systematic reviews. They are typically less structured and broader in focus. That makes them more difficult to replicate but allows the authors to show the topic through the filter of their own (subjective) lens of experience and clinical practice. A systematic review is more "technical" in nature, more focused and narrow, to exclude „bias” that, for a practicing physician seeking guidance from more experienced colleagues, is nevertheless desirable. In summary, quality narrative syntheses are more appealing to practicing clinicians, who we believe would find our paper most helpful. Therefore, we would prefer our review to remain non-systematic.

Round 2

Reviewer 2 Report

Dear authors,

A few more suggestions.

In the title (and text), please eliminate the term "Medical epicondylitis" and replace it with "Medial epicondylalgia". Both words require replacing.

Also, in the title, add the word "narrative" to describe the type of review performed.

Finally, please also include how many of the (784-12) articles you utilized for this review.

Thank you.

Author Response

Dear Reviewer

Thank you for your review and valuable comments.

Answers to specific questions and suggestions:

In the title (and text), please eliminate the term "Medical epicondylitis" and replace it with "Medial epicondylalgia". Both words require replacing.   Thank you for noticing this rather obvious mistake we made. We replaced all words with the correct version now as per your suggestion.   Also, in the title, add the word "narrative" to describe the type of review performed.   We have added that word in the title and in the 'material and methods' section.   Finally, please also include how many of the (784-12) articles you utilized for this review.   The total number of included references was 65, among which 42 were primarily focused on medial epicondylalgia. We also included that in section 2 of the manuscript.   Thank you.